# Tuning Structural Colors of TiO_2_ Thin Films Using an Electrochemical Process

**DOI:** 10.3390/molecules27154932

**Published:** 2022-08-03

**Authors:** Shumin Yang, Ao Wang, Xin Li, Guochao Shi, Yunkai Qi, Jianjun Gu

**Affiliations:** 1College of Physics and Electronic Engineering, Hebei Normal University for Nationalities, Chengde 067000, China; shumin_yang@126.com (S.Y.); wang_ao92@163.com (A.W.); lx87556@126.com (X.L.); qiyunkai@126.com (Y.Q.); 2Department of Biomedical Engineering, Chengde Medical University, Chengde 067000, China; sgc@cdmc.edu.cn

**Keywords:** TiO_2_ thin films, structural coloration, electrochemical oxidation, numerical simulation

## Abstract

TiO_2_ films exhibiting structural colors were successfully prepared using one-step electrochemical oxidation. Results of theoretical analyses and digital simulations revealed that the structural color of a TiO_2_ thin film could be regulated by adjusting oxidation voltage and oxidation time with different oxidation voltages leading to changes in structural color annulus number. At a low oxidation voltage, each thin film exhibited a single structural color, while thin films with different structural colors were obtained by varying the oxidation time. By contrast, at a higher oxidation voltage, each film exhibited iridescent and circular structural color patterns associated with symmetrical decreases in surface oxidation current density along radial lines emanating from the film center to its outer edges. TiO_2_ films exhibiting iridescent structural colorations have broad application prospects in industrial fields related to photocatalysis and photovoltaic cells.

## 1. Introduction

Colors in nature are classified as either chemical or structural colors based on the color production mechanism. Unlike chemical colors, which can only be regulated by changing molecular structures within a material, structural colors are generated through alterations of micro-nano material structures that influence reflected light interference patterns induced by diffraction or scattering of light. Therefore, the production of structural colors does not involve energy transformation, but instead relies on the structural color tuning of a single material that is achieved by adjusting periodic parameters of the ordered structure to generate multi-color output for maximal light utilization efficiency. This advantage of structural color materials has prompted researchers to actively conduct numerous studies that indicate these materials will be potentially useful as efficient, environmentally friendly, low-cost, color-fast, and easily tunable materials in diverse applications [1].

Titanium oxide (TiO_2_) is one of the most efficient semiconductor photocatalysts used in environmental applications due to its high photosensitivity, non-toxic nature, low cost, and chemical stability [2,3,4,5,6,7,8,9,10]. A new material based on this compound, TiO_2_ thin film, holds great potential as a structural color material currently under evaluation by numerous research groups testing TiO_2_ photonic crystals generated using diverse methods, including electrochemical oxidation, sol-gel methods [11], vapor deposition methods [12], liquid phase deposition [13], and magnetron sputtering [14]. Because electrochemical oxidation is a low-cost and easily controlled method, teams of researchers view this as their method of choice for generating structural color materials. For example, Zhou et al. [15] prepared two types of TiO_2_ thin films in an ethanol-glycol electrolyte, whereby the ethanol electrolyte had low electrical conductivity that generated a small ionic current that resulted in slow oxide crystal growth. In turn, this slow growth enabled the occasional formation of nanotube embryos around oxygen bubbles to build a fine film. Notably, by balancing the ionic and electron current by adjusting the proportion of glycol electrolyte, the viscosity of barrier oxide flowing around the oxygen bubbles could be regulated to influence core TiO_2_ nanotube formation. Using a different method, Zhang et al. [16] prepared TiO_2_ nanotubes based on four kinds of oxidized electric current using three different concentrations of NH_4_F electrolytes. They found that the inner diameters of TiO_2_ nanotubes did not vary markedly with NH_4_F electrolyte concentration but instead varied based on the oxygen bubble mold effect. This observation served as the basis for a new method of calculating current efficiency based on the coefficient of cubical expansion, whereby the volume expansion of oxide would significantly impact the formation of holes or nanotubes during the molding of TiO_2_ photonic crystals. Using this approach, Londhe et al. [17] prepared a TiO_2_-Al thin film, then verified that the transmissivity of this film significantly increased after it was exposed to visible light. Using an alternative method, Lou et al. [18] prepared transparent TiO_2_ thin films using a high-power pulse magnetron sputtering method. These films provided excellent performance due to their high transmittance and antibacterial characteristics, making them well-suited for use as touch screen coatings to effectively prevent surface contact-associated coronavirus transmission. Using yet another approach, Hsu et al. [19] prepared TiO_2_ thin films using atomic layer deposition to generate compact layers of solar cells to improve battery performance and reduce optical-range light energy losses due to short-wave light absorption by the TiO_2_ layer. Wang et al. [20] prepared ultrathin TiO_2_/Ti films that show single colors on the same sample by electrochemical oxidation at the oxidation voltage of 55 V and the oxidation time of 4–14 s. The thicknesses of the prepared films were from 15 nm to 52 nm, all of which were less than the critical thickness of the film interference. Moreover, the surface of the TiO_2_ film was granule, which satisfied the scattering conditions. They conclude that the color of TiO_2_ film is caused by the scattering of light. However, they did not explore the mechanism of color formation when the film thickness exceeds the critical thickness.

In conclusion, previous studies have mainly focused on preparation methods and elucidating TiO_2_ thin film formation mechanisms. However, no studies to date have investigated the impact of electrochemical oxidation efficiency during thin film preparation on TiO_2_ thin film structural color output, prompting this study. Here we employed an electrochemical oxidation method to prepare photonic crystal-like TiO_2_ thin films comprised of gradient micro-structures and then tuned structural colors of films by experimentally varying the oxidation conditions to achieve selective absorption of different wavelengths of light. Our results indicate that TiO_2_ thin films with gradient micro-structures will potentially be useful for diverse applications in industrial fields involving photocatalysis, preparation of coatings, sensors, energy storage, and color printing [21,22,23,24,25,26].

## 2. Results and Discussion

Using electrolyte mixtures containing 2.5 wt% ammonium fluoride (mass ratio) and 2% ethylene glycol mixed solution (volume ratio), two wafers were coated with TiO_2_ films via one-step oxidation using an oxidation voltage of 80 V for different amounts of time (10 s or 40 s) at room temperature. Digital images of structural colors of both TiO_2_ films (presented in Figure 1) show that the film formed via 10 s oxidation exhibited a blue color, and the film oxidized for 40 s exhibited a blue-green color. Therefore, these results suggest that as oxidation time increased, films produced were increasingly thicker, with increasing thickness leading to structural color changes.

Next, surface and cross-section morphologies of the TiO_2_ film produced via 40 s oxidation were characterized using SEM (Figure 2) to deeply explore the relationship between structural color and film microstructure. Due to uniform surface morphologies and thicknesses of separate regions within each film, only one set of SEM images is shown. In Figure 2a, closely packed holes appear on the film surface with diameters of about 10 nm and pore spacings of about 26 nm that indicate that prototypic TiO_2_ pores had formed on the titanium substrate. In Figure 2b, it can be seen that the maximum film thickness was about 130 nm. Based on an estimated oxidation rate of 3.25 nm/s, the thickness of the film generated through 10 s oxidation was calculated and found to be 32.5 nm. Using this information, the refractive index of TiO_2_ film was calculated and found to be 2.76, based on the Bragg-Snell formula:(1)$$2nd\cos\theta =(m+\frac{1}{2})\lambda$$
where *n* is the refractive index of the TiO_2_ film, *d* is the film thickness, *θ* is the diffraction angle, *m* is the interference order, and *λ* is the interference wavelength of reflected light when natural light is perpendicularly projected onto the film surface. According to Formula (1), the reflected light interference wavelength of the film oxidized for 40 s was calculated to be 478 nm, corresponding to the visible light transition from blue to green wavelengths, as consistent with the colors recorded in the digital images.

When the interference order *m* is equal to 1, and the shortest wavelength of visible light (390 nm) undergoes interference, the minimum film thickness *d* is calculated as 106 nm according to Formula (1), thus demonstrating that structural colors are not formed by optical interference involving visible light for films with thicknesses less than 106 nm. However, the 32.5-nanometer-thick film produced using 10 s oxidation appeared blue in the digital images. This blue color resulted from the absorption of yellow light (the complement of blue) and scattering of blue light by the film surface that made the film appear blue when it was irradiated with natural light projected perpendicularly onto the film surface.

Next, a series of TiO_2_ films oxidized for 35 s, 40 s, 45 s, 50 s, or 55 s were prepared using an oxidation voltage of 80 V. Digital images of these films (Figure 3) indicate that each film exhibited a single structural color, with colors ranging from purple to red with increasing film oxidation time. Calculated interference wavelengths of reflected light for these films based on SEM results using Formula (1) are shown in Table 1. It can be seen that increasing the interference wavelength of reflected light derived from a natural light source is associated with increasing oxidation time and film thickness, with predicted structural colors found to be consistent with those observed experimentally.

Next, TiO_2_ films generated via oxidation for 30 s at different oxidation voltages (90 V, 100 V, 105 V, or 110 V) were prepared to investigate the effect of oxidation voltage on structural color, with digital images of films shown in Figure 4. It can be seen from the image that the film oxidized at 90 V exhibited a single color. However, as the oxidation voltage was increased to 100 V, a red annulus appeared in the film center, with annuli increasing in size and number with increasing oxidation voltage.

To explore specific impacts of oxidation voltage changes on film microstructure, the TiO_2_ film prepared at 105 V served as a representative sample for use in investigating film surface and regional morphologies. As shown in Figure 4, film regions are marked as regions A, B, and C along the radial line from the central annulus to the outside annulus, with respective regions shown in SEM images (a), (b), and (c) in Figure 5 that reveal the correspondence between different film regions and microstructural characteristics. Since surface morphologies of all the different regions contained similar prototypic holes, only one representative image is presented here for all film regions, as shown in Figure 5d, while Table 2 presents film thickness and interference order parameters for the A, B, and C regions. It can be seen from Table 2 that film thicknesses in regions A, B, and C gradually decreased from the center to the edges from 294 nm to 259 nm to 206 nm. According to Formula (1), interference wavelengths of reflected light in regions A, B, and C were calculated as 649 nm, 572 nm, and 455 nm, respectively, which correspond to red, yellow, and blue wavelengths in the visible light spectrum, respectively, as consistent with digital imaging results.

The experimental results confirm that all other things being equal, high and low oxidation voltage effects on TiO_2_ film arise based on markedly different electrochemical oxidation mechanisms. In addition, thicknesses of different regions within each TiO_2_ film were basically the same at oxidation voltages below 90 V, leading to a single structural color (type-1 result). By contrast, thicknesses of regions of a single film decreased from the center to the edges in films formed using oxidation voltages above 100 V led to colorful, annular film structural color effects (type-2 result). Theoretically, if one beam of natural light is projected perpendicularly onto the film surface, two sub-beams reflected by the upper surface (interface between air and TiO_2_) and the lower surface (interface between TiO_2_ and Ti-base) will interfere with each other at the film surface. When the thicknesses of different film regions are the same, equal-inclination interference occurs, interference order remains the same, interference wavelengths of different regions share a constant value, and a single structural color is observed as consistent with a type-1 result. Conversely, when the thickness of each region is different, equal-thickness interference occurs, and regions with the same thickness have the same interference wavelength and thus have the same structural color, consistent with a type-2 result.

A type-1 result indicates the existence of a uniform electric field between the anode and the cathode during electrochemical oxidation that supports the same oxidation current density across all film surface regions, with current line distributions for this process shown in Figure 6a. By contrast, a type-2 result indicates that a non-uniform electric field exists between the anode and the cathode during the electrochemical process, such that the oxidation current density gradually decreases from the center of the film but is mainly the same throughout every single annulus. Based on the experimental results, it could reasonably be assumed that lines of electrochemical reaction current at higher oxidation voltages are radiating points, as shown in Figure 6b. Since the thickness of the film gradually became thinner as the current density of the point-like electrode gradually decreased along the radial line emanating outward from the film center, an annular and iridescent structural color was observed due to equal-thickness interference that occurred under natural light, as previously reported [27]. In addition, a region of color transition was observed between adjacent annulus colors due to the superposition of reflected light interference wavelengths in adjacent micro-areas with continuously changing micro-thicknesses.

As an additional assessment, we measured peak reflectivities of TiO_2_ film oxidized at 105 V using spectral tests, with the results shown in Figure 7. It can be seen from the figure that wavelengths of peak reflectivities were 455, 571, and 717 nm in visible light, which corresponded to colors blue, yellow and red, respectively, as consistent with colors observed in digital images of the film.

Concerning current density effects as reported in the literature [27], when a carbon rod is equivalent to a point electrode, the formula for calculating the current density on a film surface is:(2)$$J=k\frac{Uh}{{\left(r^{2}+h^{2}\right)}^{\frac{3}{2}}}$$
where *k* is a constant related to the electrolyte and temperature (in units of Ω^−1^), *r* is the distance from the film center to a certain point on the surface (*r* < 0.7 cm), h is the distance from the negative carbon to the anode (about 6 cm), and *U* is the oxidation voltage. Numerical simulations of the current density *J* of the anodizing film shown in Figure 4 were carried out, and current density distributions for different oxidation voltages are shown in Figure 8. It can be seen from the figures that the current density of the anodizing film also increases with increasing oxidation voltage, such that for a given oxidation voltage, the current density at the center of the film center is greatest and decreases symmetrically outward in all directions from the center. Thus, the track with the same current density is circular and thus appears as a single annular structural color.

Figure 9 shows the correlation curve between the current density gradient and oxidation voltage for analyzing the relationship between oxidation voltage and structural color annulus number. It can be seen from the figure that the current density gradient along a radial going outward from the center also increases with increasing oxidation voltage leading to the development of a gradient of increasing thickness along each radial in the same direction. When the optical path length difference of reflected light of upper and lower surfaces exceeds a certain wavelength range, the number of structural color annuli formed through equal-thickness interference increases. Ultimately, film generated using an oxidation voltage of 90 V had a lower current density gradient along the radial direction that resulted in an optical path difference from center to edge that fell within a limited wavelength range, resulting in a lack of obvious annular structural color.

## 3. Experimental

A high-purity titanium sheet (99.99%) of 2 mm thickness was cut to form a round wafer with a diameter of about 2 cm. The flat, round wafer was dipped in acetone, then alcohol, and then the wafer was ultrasonically cleaned for 10 min, rinsed repeatedly with deionized water, and allowed to dry. Thereafter, the dry wafer was dipped in a polishing solution containing hydrofluoric acid (40%), concentrated nitric acid (65%), and deionized water (combined 1:4:5 by volume) for 2 min. After polishing, the wafer was rinsed with deionized water again, dried, and stored for later use.

A pre-assembled electrolyzer was used for TiO_2_ thin film preparation via electrochemical oxidation. First, a pretreated titanium wafer and a carbon rod serving as anode and cathode, respectively, were arranged in parallel with a 6 cm gap between them. Electrochemical oxidation was performed using an electrolyte that contained a mixture of 2.5 wt% ammonium fluoride and 2% ethylene glycol solution (volume ratio), with effective oxidation voltages ranging from 80–110 V and oxidation times ranging from 10 s to 55 s. The electrochemical oxidation device used to prepare structural color materials based on a series of TiO_2_ thin films is shown in Figure 10.

A digital camera (Canon EOS600D) was used to record the structural colors of TiO_2_ thin films then a scanning electron microscope (SEM, Zeiss Sigma-300) was employed to analyze the microstructural characteristics of different areas of each film. Thereafter, the visible reflection spectrum of each film sample was determined using an ultraviolet-visible spectrophotometer (Hitachi U-3010). In addition, Matlab software was used to simulate the current density distribution of each TiO_2_ thin film.

## 4. Conclusions

We prepared TiO_2_ films for use as structural color materials using electrochemical anodization. The results of our experiments revealed that for films with a thickness of less than 106 nm, the scattering of light by the films was responsible for their structural colors. By contrast, when film thicknesses exceeded 106 nm, the structural color originated from interference between light reflected by film upper and lower surfaces. When the oxidation voltage was less than 90 V, each film exhibited a single structural color that could be tuned by changing the oxidation time. When the oxidation voltage was greater than 100 V, the film exhibited gradient and iridescent structural color annuli that increased in number with increasing oxidation voltage. Theoretical studies and numerical simulations revealed that high and low oxidation voltages could induce structural color changes based on different electrochemical oxidation mechanisms. At higher oxidation voltages, film thickness and film surface oxidation current density gradually and radially decreased from the center outward along radiating current lines. In addition, the current density gradient and film thickness gradient also increased radially from the center outward due to increased oxidation voltage, which ultimately generated increasing numbers of structural color annuli as the distance from the film’s center increased. Taken together, these findings provide valuable information to guide the development of TiO_2_ thin films for use in various industrial applications.

## Figures and Tables

**Figure 1 molecules-27-04932-f001:**
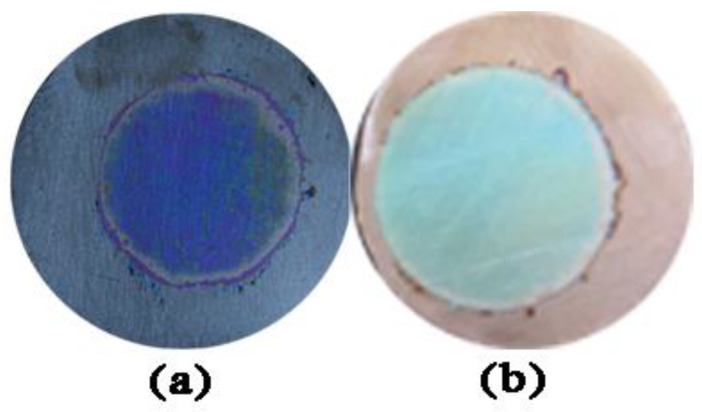
Digital images of TiO_2_ films oxidized with a voltage of 80 V for (**a**) 10 s and (**b**) 40 s.

**Figure 2 molecules-27-04932-f002:**
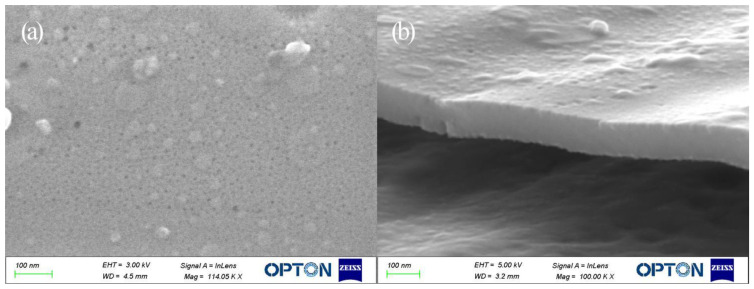
SEM surface and cross-sectional images of TiO_2_ film oxidized with a voltage of 80 V for 40 s. (**a**) Surface, (**b**) Cross section.

**Figure 3 molecules-27-04932-f003:**
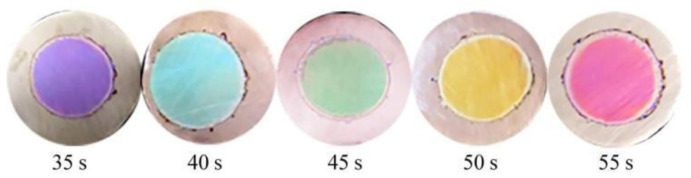
Digital images of TiO_2_ films oxidized with a voltage of 80 V for 35, 40, 45, 50, and 55 s.

**Figure 4 molecules-27-04932-f004:**
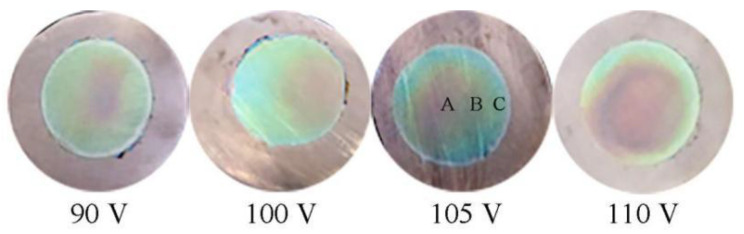
Digital images of TiO_2_ films oxidized with voltages of 90, 100, 105, and 110 V for 30 s.

**Figure 5 molecules-27-04932-f005:**
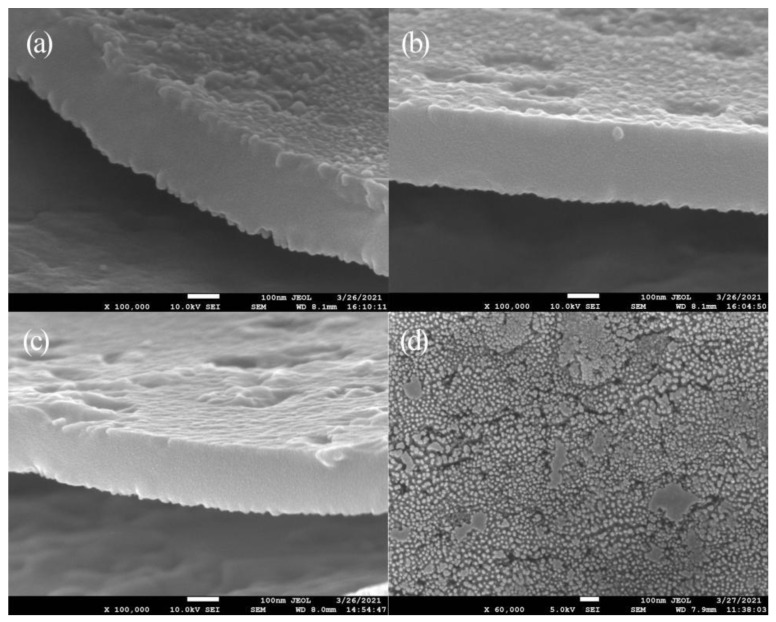
SEM surface and cross-sectional images of TiO_2_ film oxidized with a voltage of 105 V for 30 s. Parts (**a**–**c**) correspond to positions A to C in Figure 4. Part (**d**) corresponds to position A in Figure 4.

**Figure 6 molecules-27-04932-f006:**
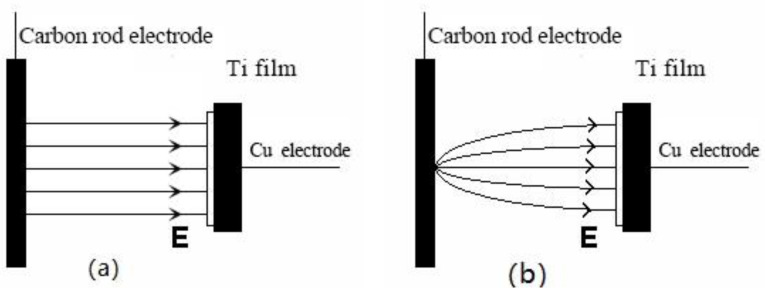
Schematic diagram of power lines of different oxidation voltages. (**a**) Low voltage, (**b**) High voltage.

**Figure 7 molecules-27-04932-f007:**
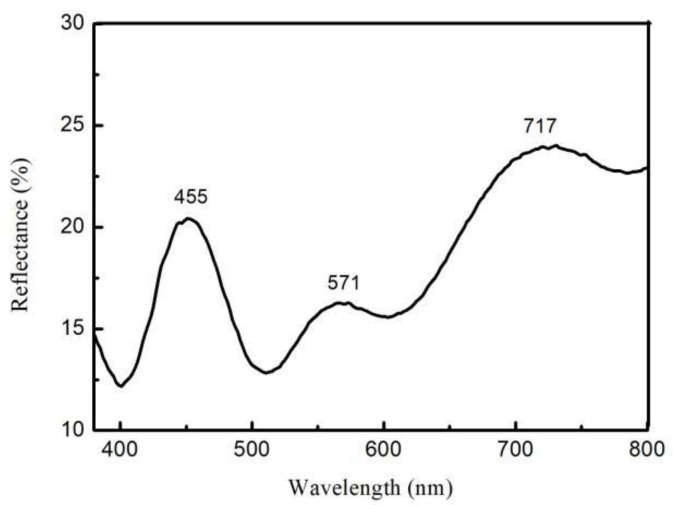
The reflection spectrum of the TiO_2_ film oxidized with 105 V for 30 s.

**Figure 8 molecules-27-04932-f008:**
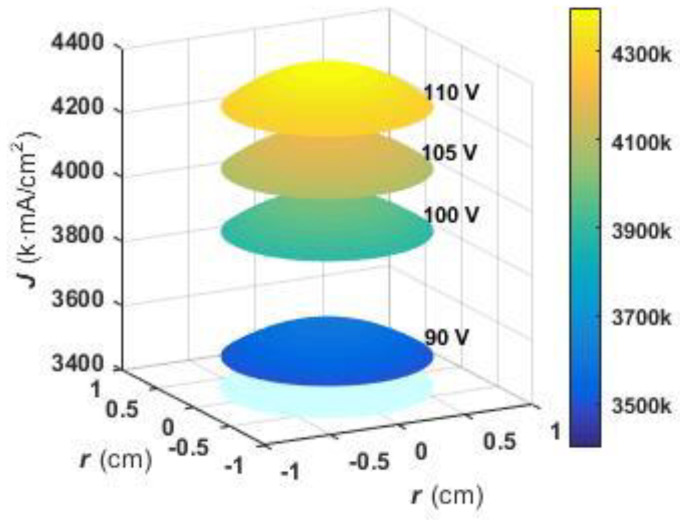
Simulation current density of TiO_2_ films oxidized with different voltages.

**Figure 9 molecules-27-04932-f009:**
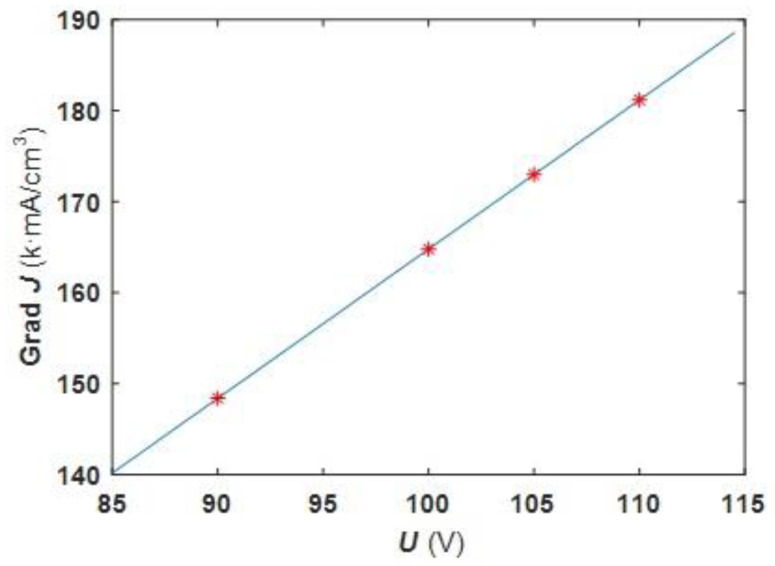
Curve of average current density gradient as a function of oxidation voltage.

**Figure 10 molecules-27-04932-f010:**
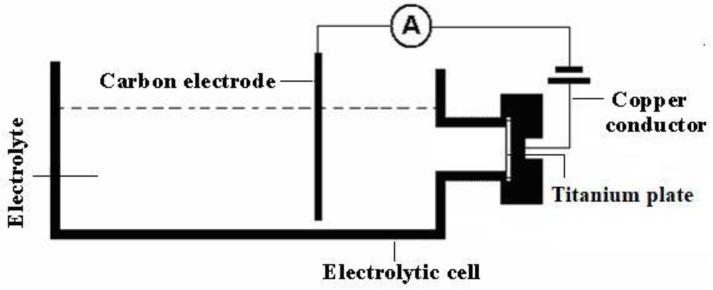
Schematic diagram of electrochemical oxidation device.

**Table 1 molecules-27-04932-t001:** The parameters of TiO_2_ thin films are shown in Figure 3.

Oxidation Time (s)	35	40	45	50	55
Thickness of film (nm)	114	130	146	163	179
Interference wavelength (nm)	419	478	538	600	658
Interference level	1	1	1	1	1
Structural color	purple	turquoise	green	orange	red

**Table 2 molecules-27-04932-t002:** The parameters of different regions in TiO_2_ thin film shown in Figure 4.

Regions	A	B	C
Thickness of film (nm)	294	259	206
Interference wavelength (nm)	649	572	455
Interference level	2	2	2
Structural color	red	yellow	blue

## Data Availability

Data sharing does not apply to this article.

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
