# Peer review of "Tuning Structural Colors of TiO2 Thin Films Using an Electrochemical Process"

_molecules, 2022, doi:10.3390/molecules27154932_

Round 1

Reviewer 1 Report

The manuscript describes the preparation of TiO2 films by one-step electrochemical oxidation, exhibiting different structural colors depending from the preparation conditions. The subject is interest and well presented but a similar paper entitled “Synthesis of ultrathin TiO2/Ti films with tunable structural color” by Yanlu Wang, Rushuai Han, Liqian Qi, Lihu Liu, and Huiyuan Sun, has been published at Applied Optics Vol. 55, Issue 35, pp. 10002-10005 (2016)  https://doi.org/10.1364/AO.55.010002.

Strangelly there is no reference to this paper although in both cases the authors are from the same city and probably from the same University.

For this reason, I propose to add the previous ref, comment their results in comparison with the previous results and if there are any new results or conclusions to be accepted for publication.

Reviewer 2 Report

The manuscript entitled "Tuning structural colors of TiO2 thin films using an electro- 2 chemical process" by Yang et.al. reports the way to tune the optical color of TiO2 depending on deposition process conditions. This will be useful for diverse applications in industrial fields involving photocatalysis,  preparation of coatings, sensors, energy storage and color printing. I would like to recommend this manuscript to be published in molecules after considering the below comments.

There are many reasons to show the different colors. More specific evidence is needed. For example, crystalline size (by XRD), surface roughness (AFM), chemical composition changes (XPS), or show the optical band-diagram (based on UV-Vis and XPS data (valence band offset values). More specific scientific evidences are critically needed to explain the mechnisms.

Round 2

Reviewer 1 Report

After the last additions concerning the comments about the previous work, the innovative character is more clear. 

Reviewer 2 Report

Addressed well